# Integrating Immunotherapy into Multimodal Treatment of Head and Neck Cancer

**DOI:** 10.3390/cancers15030672

**Published:** 2023-01-21

**Authors:** Yuan James Rao, Joseph F. Goodman, Faysal Haroun, Julie E. Bauman

**Affiliations:** 1Division of Radiation Oncology, The George Washington University School of Medicine, Washington, DC 20037, USA; 2Division of Head and Neck Surgery, The George Washington University School of Medicine, Washington, DC 20037, USA; 3Division of Hematology/Oncology, The George Washington University School of Medicine, Washington, DC 20037, USA

**Keywords:** immunotherapy, immune checkpoint inhibitors, head and neck squamous cell carcinoma, radiation, surgery, chemotherapy, immunochemoradiotherapy

## Abstract

**Simple Summary:**

Immune checkpoint inhibitors have become the standard of care therapy in a multimodal setting for recurrent/metastatic HNSCC. Multiple clinical trials have recently looked at the addition of ICI to multimodal treatments in locally advanced HNSCC. Multiple Phase II/III trials are investigating the combination of ICI with definitive chemoradiation. Phase I/II trials have concluded that neoadjuvant ICIs are relatively safe when given prior to surgery and do not generally cause a delay in proceeding to surgery within 4 to 6 weeks. A significant pathological response occurs in about 20% of cases with monotherapy and may be higher with combination therapy. Phase III trials are ongoing to include neoadjuvant immunotherapy along with adjuvant immunotherapy for high-risk features in the postoperative setting along with chemoradiation.

**Abstract:**

Patients with locally advanced head and neck squamous cell carcinoma (HNSCC) have a poor prognosis, with a significant risk of progression or death despite multimodal treatment with surgery, chemotherapy, and radiotherapy. Immune checkpoint inhibitors targeting the programmed death receptor-1 (PD1) have dramatically changed the treatment landscape for recurrent/metastatic disease, improving overall survival in both the first- and second-line palliative settings. This success has driven the investigation of treatment strategies incorporating immunotherapy earlier into the multimodal curative-intent or salvage treatment of both locally advanced and recurrent/metastatic HNSCC. This review encompassed the following three subjects, with a focus on recently reported and ongoing clinical trials: (1) the use of neoadjuvant immunotherapy prior to surgery for locally advanced HNSCC, (2) the use of immunochemoradiotherapy for locally advanced head and neck cancers, and (3) novel uses of immunotherapy in the salvage of recurrent/metastatic HNSCC via a combined modality, including reirradiation paradigms. The results of these studies are eagerly awaited to improve patient outcomes in this challenging disease.

## 1. Introduction

Patients with locally advanced Stage III–IVb head and neck squamous cell carcinoma (HNSCC) have a poor prognosis, with significant risk of progression or death despite modern multimodal treatment with surgery, chemotherapy, and radiation therapy (RT). The long-term MACH-NC meta-analysis of 19,805 patients with locally advanced HNSCC in 107 historical randomized trials reported that fewer than 50% of patients survived for 5 years, despite multimodal curative-intent treatment [1]. Immune checkpoint inhibitors (ICIs), specifically monoclonal antibodies (mAbs) against programmed death receptor-1 (PD-1), have dramatically changed the treatment landscape for HNSCC. Nivolumab and pembrolizumab, both anti-PD1 mAbs, were shown to improve overall survival (OS) over the standard single-agent therapy in the second-line treatment of platinum-refractory, recurrent/metastatic HNSCC; in 2019, pembrolizumab, either alone or in combination with chemotherapy, was shown to be superior to the standard therapy in first-line treatment [2,3,4].

These recent advances in treatment of recurrent/metastatic HNSCC have led to the rapid development of clinical trials to test strategies incorporating immunotherapy earlier into the multimodal curative-intent or salvage treatment of HNSCC. This review encompassed the following three subjects, with a focus on recently reported and ongoing clinical trials: (1) the use of neoadjuvant immunotherapy prior to surgery for locally advanced HNSCC, (2) the use of immunochemoradiotherapy for locally advanced HNSCC, and (3) novel uses of immunotherapy in the salvage of recurrent/metastatic HNSCC via a combined modality.

## 2. The Immune Microenvironment in HNSCC

The tumor microenvironment (TME) in HNSCC is complex, with several microscopic niches [5]. First, the invasive front/perivascular niche includes the tumor edge next to the normal epithelium and next to the vasculature. This niche typically contains the cancer stem cells and the most highly proliferative tumor cells. The interior of the cancer consists of the central tumor compartment, and contains cancer cells that are less proliferative and have a glycolytic metabolism due to hypoxia. Additional non-cancer cells in the TME can include cancer-associated fibroblasts, which arise from the population of circulating fibroblasts and co-evolve with the tumor into a distinct phenotype that is involved in carcinogenesis. Cancer-associated fibroblasts may express cytokines such as hepatocyte growth factor, CXCL12, and TGF-beta, which promote invasion and angiogenesis. The interaction between fibroblasts and the tumor creates a stromal environment that also may inhibit the ability of immune cells and cancer treatments to penetrate into the tumor [6].

Importantly for the purposes of this review, immune cells are also present in the TME. However, the persistent unresolved inflammation associated with cancer results in the eventual decay and malfunction of the normal immune processes, which allows for continued progression and growth of the squamous cell carcinoma component of the TME. The most important cell type of the immune TME are the T-lymphocytes, which regulate the adaptive immune response and elicit a cytotoxic response to tumors. Tumor-infiltrating T cells are present in many HNSCC but are dysfunctional, indicating that tumors suppress this component of the TME. Specific functional deficits of T cells within HNSCC may include: (1) downregulation of signaling receptors, (2) decreased proliferation, (3) the inability to kill tumor cells, (4) imbalance in the cytokine profile, and (5) the initiation of T cell apoptosis [7]. Additional factors that may affect the immune environment include circulating regulatory T cells (Tregs) that suppress the activity of cytotoxic T cells, the upregulation of immune checkpoint ligands such as CTLA-4 and PD-L1, and the defective function of antigen-presenting cells (APC) [8,9]. All of these factors lead to an immune reaction that is insufficient for suppressing the growth of squamous cell carcinomas in the TME.

The pre-existing immunosuppressive effects of the tumor itself can be exacerbated by treatments including surgery, chemotherapy, and radiation. Surgery necessarily results in substantial tissue and vascular disruption, and surgery-induced necrotic cell death leads to a release of a number of sequestered cellular factors such as growth factors, clotting factors, stress hormones, and cytokines [10]. Through these mechanisms, surgery leads to immune suppression that peaks at 3 days and may last for several weeks. Additionally, immune cells are highly sensitive to radiation, and leukocytes typically undergo apoptosis when exposed to radiation doses of an order of magnitude less than those used for the therapeutic treatment of squamous cell carcinomas [11]. Therefore, radiotherapy also causes lymphopenia in more than 75% of patients with HNSCC, and their absolute lymphocyte counts may remain depressed for up to a year after treatment [12]. Chemotherapy does not discriminate in clearing immune cells primed against cancer or those involved in ongoing immune suppression, and lymphopenia is a common side effect of chemotherapy used as a treatment for head and neck cancer as well.

The introduction of immune checkpoint inhibitors to this complex microenvironment, which is perturbed by these other treatments, introduces several opportunities. The main principle is that the existing tumor may be used as an “in-situ” tumor vaccine and a source of antigens for dendritic cell antigen presentation to activate cytotoxic T cells [13]. Checkpoint inhibition by ICIs may overcome T cell dysfunction and allow the immune system to recognize the tumor neoantigens and suppress cancer in the local tumor. Importantly, activation of the immune system might also result in the elimination of subclinical distant metastatic disease and, hypothetically, could improve survival by preventing the development of clinically apparent metastatic disease. The timing of immunotherapy in relation to the other cancer treatments may be significant as well, in that the initial suppressive immune microenvironment after surgery can be replaced via cytoreductive therapies (such as radiation or chemotherapy) and enhanced with immunotherapy after adequate immune recovery [14]. Maintenance of the normally draining lymph nodes may also be critical for re-establishing the immune surveillance of tumors [15]. Given that there are many ways that immunotherapy can be combined and sequenced with surgery, radiation, and chemotherapy, it is important to conduct clinical trials to identify combinations in which the addition of immunotherapy may be superior to the standard treatment. In the following sections, we will discuss the significant work being done in ongoing in phase I/II/III trials to determine the best integration of multimodal treatments into this complex immune environment.

## 3. Immunotherapy in Combination with Surgery for Locally Advanced HNSCC

The earliest descriptions of immunotherapy and surgery were found in the Egyptian era, circa 1550 BC. The Ebers papyrus describes the application of a poultice after the incision of a tumor, leading to infection and tumor regression. Lore and legend is rich with examples of the “spontaneous regression” of tumors, often associated with the infection of a tumor and its subsequent clearance, as in the case of Saint Peregrine, patron saint of cancer [16]. In the modern era, new interest in the infectious treatment of cancer began with a report in the German literature in 1883 by Friedrich Fehleisen describing the regression of a fibrosarcoma of the cheek after infection with *Streptococcus (erysipelas) pyogenes* [17]. Dr. William Coley of New York injected an inoperable sarcoma of the tonsil and neck with “5 decigrams of a bouillon culture of streptococcus” in 1891, with complete regression of disease noted within 2 weeks after a high fever and “a typical attack of erysipelas” [18]. Before the current era of anti-PD1 mAbs and an understanding of the diverse population of cytotoxic, helper, and regulatory tumor-infiltrating lymphocytes (TIL), intriguing results with vaccines and interferons occasionally produced dramatic results in some cancers (e.g., melanoma) [19]; however, these results were often inconsistent, and no clear immune mechanism was understood.

The effects of surgery on the immune system are understood in much greater detail now. The role of APCs such as dendritic cells, macrophage ratios, and Foxp3+ Tregs continue to be studied regarding their effects on immune regulation, which may allow some cancers to persist. Surgery can disrupt this delicate balance, allowing new immune mechanisms of surveillance and attack to clear cancer. On the other hand, the cytokine response after local tissue injury induced by surgery can have immunosuppressive effects [20]. The concepts of immune escape and tolerance are especially relevant to the mucosa of the upper aerodigestive tract, which in the normal state does not exhibit significant chronic inflammation despite constant exposure to allergens, bacteria, and other foreign antigens [21].

After surgery, the immunosuppressive environment has been shown to be enhanced by increased expression of interleukins including IL-4, IL-6, and IL-10, as well as the downregulation of interferon gamma and IL-2. Factors contributing to the systemic changes include extensive surgery, anesthesia, and residual tumors [22]. Some of these factors normalize after about 2 weeks; however, the long-term effects are not clear. The data suggest that antitumor immunity can be developed during a short timeframe after removal of the tumor and the associated suppressive microenvironment while some of the tumor antigens remain. On the other hand, during standard treatments, the presence of a grossly positive margin portends a poor prognosis, even when ultimately cleared and followed by CRT [23]. The key concepts when combining immunotherapy and surgery during primary treatment would be both reducing the tumor immune-suppressive characteristics related to the tumor’s volume and a suppressive immune microenvironment (myeloid-derived suppressor cells, Tregs, macrophages, etc.).

## 4. Neoadjuvant Immune Checkpoint Inhibitor (ICI) Trials

In the past 5 years, clinical trials studying a combination of surgery and immunotherapy have increased by nearly fivefold. In the case of HNSCC, most current studies are focused on neoadjuvant immunotherapy with anti-PD1 mAbs. Although anti-PD1 mAbs in the recurrent/metastatic setting increase overall survival, durable effects are observed in only about 20% of cases. In principle, people with untreated, operable HNSCC are less immunosuppressed than those with recurrent/metastatic disease; administering an anti-PD1 ICI in the neoadjuvant setting may allow a more robust immune response. A recent systematic review [24] reported over 20 trials currently ongoing for neoadjuvant immunotherapy prior to surgery for mucosal HNSCC. Specifically, all trials included the use of neoadjuvant ICIs including nivolumab, pembrolizumab, ipilimumab, durvalumab, and tremelimumab. Eight clinical trials met the inclusion criteria for analysis, with a total of 260 patients, showing an overall objective response rate of about 40%, which was slightly better with combination therapy, and with no deaths or immune-related toxicities. Several key trials are highlighted below and summarized in Table 1.

Several clinical trials have used the concept of administering immunotherapy during the “window of opportunity” that occurs between biopsy-proven diagnosis and the definitive surgical procedure. This window inevitably occurs because of the time (usually several weeks) needed to perform multidisciplinary consultations and staging imaging, and to schedule the definitive surgery. These studies rationally inserted immunotherapy into this time, in which the patient would not otherwise be receiving cancer-directed treatment. These study designs are useful for measuring the efficacy of immunotherapy in patients with localized disease and are also useful for sampling paired pre- and post-treatment biomarkers of immune effects, since the tumor is fully resected.

Three published studies have evaluated single-agent monotherapy as a neoadjuvant immunotherapy therapy prior to surgery. CheckMate-358 [25] was a Phase I/II trial comparing previously untreated, resectable HPV-positive or HPV-negative HNSCC treated with nivolumab in two doses two weeks apart prior to surgery. The use of nivolumab in the neoadjuvant setting was determined to be generally safe and well tolerated, with a low risk of high-grade adverse events. Pathologic regression of tumor was seen in 23.5% of HPV-positive tumors vs. 5.9% of HPV-negative tumors. A neoadjuvant Phase II trial of pembrolizumab for locally advanced HPV-negative HNSCC [26] enrolled 36 patients receiving one preoperative dose of the checkpoint inhibitor, followed by surgery within 3 weeks. A pathologic response rate of 44% was observed, which was predictive of better recurrence-free survival after postoperative chemoradiation. The overall risk of relapse at 1 year was reduced compared with historical controls. Tumor downstaging was observed in 19% of patients. In a separate Phase II trial, pembrolizumab was used in 75 patients with HPV-negative T3/T4 HNSCC and more than two lymph nodes [27]; 43% of the patients demonstrated a treatment effect of >20%, and the responders had 93% DFS.

Additionally, three published studies have tested multiagent immunotherapy prior to surgery. The IMCISION trial combined nivolumab with ipilimumab to study the pathologic tumor response prior to surgery in HNSCC at Stage II or greater. In this trial, 30% showed near pCR and 60% showed some response. RFS was 100% in near pCR at 14 months [28]. In a separate Phase II trial, nivolumab was used with or without ipilimumab prior to surgery for T2N1 (or greater) oral cavity cancers in 29 patients, showing a 73% response rate to the combination treatment, which was defined as a pathological response (tumor necrosis/histiocyte reaction) of at least 10%. About 70% treated with the monotherapy had downstaging, which allowed de-escalation of the adjuvant treatment [29]. The CIAO trial evaluated durvalumab (anti-PD-L1) with or without tremelimumab (anti-CTLA-4) in 28 patients prior to surgery, with 29% showing tumor reduction to 10% of the pretreatment volume and 25% showing downstaging of the tumor [30].

These preliminary studies of single or multiagent immunotherapy prior to surgery demonstrated that overall, the treatment approach was well tolerated, with response rates to immunotherapy observed to be in the approximate range of 40–70% of patients, tumor downstaging was observed in approximately 20–25% of patients, and significant pathological responses were seen in approximately 20% of patients. Patients who demonstrated a good tumor response to immunotherapy generally had a favorable prognosis after surgery. Although it is difficult to draw comparisons among these small studies, combination immunotherapy may have a numerically higher response rate than monotherapy.

These initial promising results motivated the development of trials that have combined immunotherapy with either radiotherapy or chemotherapy. A single-institution Phase Ib clinical trial in Oregon (NCT03247712) treated 21 patients with neoadjuvant stereotactic body radiation (SBRT, either 40 Gy in five fractions or 24 Gy in three fractions) to the primary tumor only, with or without nivolumab, prior to definitive surgical resection for HPV-positive/negative HNSCC [31]. The treatment was determined to be safe, with a robust tissue response and clinical downstaging in 90% of patients. A major pathologic response was seen in 86% and a complete pathologic response was seen in 67%. A single-institution Phase 2 trial at Thomas Jefferson University (NCT03342911) used nivolumab along with carboplatin and paclitaxel, given weekly for 6 weeks prior to surgery for Stage III/IV HPV-negative or Stage II/III HPV-positive HNSCC. The initial data presented at ASCO in 2020 [32] showed good tolerance of the regimen without significant adverse events, with a pathologic complete response (pCR) seen in about 40% regardless of the HPV status. This study built upon the work by Sadeghi and Siegel [33], showing significant pCR using cisplatin and docetaxel in the neoadjuvant setting for HPV-positive OPSCC prior to trans-oral robotic surgery (TORS). The Morpheus—Head and Neck Cancer Trial (NCT05459129) is an impressive four-arm comparison of atezolizumab in the neoadjuvant setting before surgery in LAHNSCC. This is a Phase Ib/II multicenter study of resectable p16-negative HNSCC. The arms include anti-PDL1 monotherapy with atezolizumab (two cycles over 6 weeks prior surgery), atezolizumab combined with tiragolumab (a novel anti-TIGIT monoclonal antibody) 6 weeks prior to surgery, atezolizumab/tiragolumab with SBRT prior to surgery; and a neoadjuvant atezolizumab–tiragolumab combination along with carboplatin/paclitaxel neoadjuvant chemotherapy prior to surgery. This study seeks to enroll 180 patients over 3 years. The abovementioned studies will determine whether multimodal neoadjuvant treatment with immunotherapy plus SBRT or chemotherapy should be tested in future Phase III studies.

In addition to immune checkpoint inhibitors, several other agents are being investigated as neoadjuvant therapy prior to surgery. The anti-KIR mAb lirilumab, which stimulates natural killer (NK) cells by preventing KIR–HLA interactions, was combined with nivolumab in a neoadjuvant Phase II study of 28 patients with recurrent HNSCC prior to salvage surgery [34]. The head and neck cancer subsites included the oral cavity, oropharynx, and larynx/hypopharynx (even distribution), and most had undergone prior radiation. A pathological response was seen in 43%. The reported 1-year DFS was 55.2% and the OS was 85.7%; the 2-year DFS and OS were 64% and 80% in responders. A Phase II trial of a live-attenuated *Listeria monocytogenes* vaccine [35] encoding an HPV16 E7 oncoprotein has been studied in a window trial prior to TORS (ADXS11-001); however, because of safety concerns, including one death, this and several similar trials using axalimogene filolisbac (AXAL) have been terminated or suspended [36]. A Phase Ib/II vaccine trial (MEDI0457/INO-3112) used synthetic DNA plasmids targeting E6/E7 antigens with recombinant IL-12 given before and after surgery for HPV-positive HNSCC. The treatment requires electroporation with the CELLECTRA device. The induction of HPV-specific CD8+ T cell immunity was reported; however, only two patients completed the full treatment in Cohort 1 before and after surgery [37]. While promising, these novel immunotherapy methods require further research before advancing to Phase III testing.

Finally, the concept of presurgical immunotherapy with checkpoint inhibitors is currently being tested in one notable Phase III study. Keynote-689 (NCT03765918) is an ongoing randomized Phase III trial that evaluates pembrolizumab in the neoadjuvant setting for locally advanced resectable head and neck cancer. Patients will be randomized to receive the standard surgical resection and adjuvant therapy vs. the investigational treatment, which will include neoadjuvant pembrolizumab for two cycles followed by surgical resection then the standard adjuvant treatment plus adjuvant pembrolizumab (15 cycles). The study began enrolling worldwide in 2018 and has an enrollment target of over 700 patients. No preliminary data have yet been published, but these are eagerly awaited to determine if neoadjuvant immunotherapy should be added to the standard of care for patients with LAHNSCC suitable for surgery.

At the present time, it is premature to predict if neoadjuvant ICI will become the standard of care for resectable, locally advanced HNSCC, but this may change if the Keynote-689 trial reports a positive result. However, the studies mentioned above have shown that neoadjuvant ICI is well tolerated, oncologically safe, and with reasonable rates of tumor response and downstaging. Future directions for neoadjuvant ICI trials may include treatments combining ICI with SBRT, chemotherapy, or other immunomodulating systemic agents. The identification of biomarkers for the selection of the patients who are most likely to respond to neoadjuvant ICI will also be important. Candidate biomarkers include PD-L1 expression, the tumor’s mutational burden, HPV status, and certain immune phenotypes such as high CD8+ T cell infiltration [21].

## 5. Immunochemoradiotherapy in Locally Advanced HNSCC

The standard non-surgical management of locally advanced HNSCC is curative-intent chemoradiotherapy (CRT), typically involving RT doses of around 70 Gy in 35 fractions in combination with concurrent cisplatin-based chemotherapy. Although it has been recognized as the standard of care for more than two decades, CRT has suboptimal efficacy, especially in human papillomavirus (HPV)-negative or Stage III HPV-associated disease. Efforts to improve the efficacy of definitive CRT by altering the RT fractionation or adding cetuximab, an anti-epidermal growth factor receptor mAb, have not been successful [38,39].

In theory, the addition of ICIs to the backbone CRT regimen, a treatment strategy called “immunochemoradiotherapy”, could improve the oncologic outcomes. CRT may facilitate immunogenic cell death and antigen release from the tumor, thereby enhancing the immune response to ICI and facilitating long-term immune surveillance to reduce both local and distant recurrences [40]. JAVELIN H&N 100 was a Phase III trial of 907 patients with locally advanced HNSCC who were randomized to standard CRT vs. immunochemoradiotherapy with the addition of avelumab, an anti-PD-L1 mAb [41]. Patients were unselected for PD-L1 and stratified by HPV status. Avelumab was given concurrently with CRT and as maintenance therapy for up to 12 months. The primary endpoint of improved progression-free survival (PFS) was not met, although immunotherapy did not increase the risk of serious treatment complications. KEYNOTE-412 was a Phase III study of 780 patients with locally advanced HNSCC randomized to standard CRT vs. immunochemoradiotherapy with the addition of concurrent and adjuvant pembrolizumab (NCT03040999). This combination also failed to improve the primary endpoint of event-free survival [42]. Additionally, GORTEC-REACH (NCT02999087) was a Phase III trial that randomized 430 cisplatin-eligible patients with locally advanced HNSCC to standard CRT vs. RT with avelumab and cetuximab. The 1 year progression-free survival was reported to be 73% with CRT and 64% with the experimental treatment, crossing the futility boundary and favoring the standard of care [43]. To our knowledge, no other large Phase III studies are testing the concept of concurrent immunochemoradiotherapy in unresected, locally advanced HNSCC, and this developmental strategy has largely been abandoned.

A substantial portion of patients with locally advanced HNSCC are considered to be unsuitable to receive the standard of care, namely cisplatin-based CRT, for a variety of factors including performance status, age, comorbidities, and other factors. In this situation, for cisplatin-ineligible patients, the current standard of care is to treat them with cetuximab and radiation rather than radiation alone, but these patients are generally considered to have a poor prognosis, and immunotherapy is being investigated to see if their treatment outcomes can be improved [44]. The GORTEC-REACH trial also enrolled 275 cisplatin-ineligible patients and randomized them to standard treatment with cetuximab and RT vs. the standard treatment plus avelumab. The 2-year PFS was numerically higher in the experimental arm at 44% vs. 31% for the standard of care, but the difference was not statistically significant [43]. NRG-HN004 (NCT03258554) randomized 186 patients to receive either the standard of care, which was cetuximab and RT vs. duvalumab and RT. Recently, the 2-year PFS rates were reported to be 51% for the durvalumab arm vs. 66% for the cetuximab arm, which did not meet the primary endpoint [45].

In contrast to the previously discussed disappointing studies of definitive immunochemoradiotherapy for HNSCC, the addition of immunotherapy to standard CRT has been shown to improve survival in locally advanced non-small cell lung cancer (NSCLC) and esophageal cancer. In the PACIFIC trial, the addition of adjuvant durvalumab, an anti-PD-L1 mAb, to definitive CRT for Stage III NSCLC resulted in improved survival compared with standard CRT [46]; thus, immunochemoradiotherapy has become a standard of care for NSCLC [47]. In Checkmate 577, the addition of adjuvant nivolumab to the standard of care (CRT and surgery) for esophageal cancer also resulted in improved survival [48]. We hypothesize that two possible factors may explain the discrepant results. Firstly, the elective nodal radiotherapy in HNSCC might decrease the benefit of immunotherapy; second, adjuvant immunotherapy may be a more effective strategy than concurrent immunotherapy with RT. Elective nodal radiation is commonly used in head and neck RT and treats a large volume of lymph nodes that potentially harbor subclinical disease with moderate doses in the range of 50–60 Gy. Although potentially improving the rates of locoregional nodal control, elective nodal RT can increase the rates of lymphopenia and immune suppression by eradicating regional lymphocytes [49]. Elective nodal RT may decrease the benefit of immunochemoradiotherapy in HNSCC by inhibiting T cell recruitment and priming, or by killing effector T cells within the immune microenvironment. It is notable that the Phase III CALLA trial (NCT03830866) for locally advanced uterine cervix cancer, in which definitive radiotherapy with elective nodal radiation is commonly practiced, also reported that the addition of durvalumab did not improve PFS over chemoradiation alone [50]. In contrast, elective nodal radiation is not widely used in NSCLC and esophageal cancer, which could explain the positive results of the PACIFIC trial and Checkmate 577. Neither trial specified the design of the radiation fields nor reported on the use of elective nodal radiation in their publications, and our hypothesis is based on the common practice of not treating the elective nodes in these disease sites [51,52,53].

Strategies to reduce the irradiated volume may also increase the efficacy of immunotherapy by minimizing the destruction of the responding immune cells. Such strategies, also known as “volume de-escalation”, have previously been investigated with the goal of reducing the toxicity of head and neck radiotherapy. Although wide elective nodal radiation volumes were once commonly used, it has become more common to selectively decrease or omit the elective nodal radiation volumes in the high Level II and retropharyngeal lymph nodes in oropharyngeal cancer [54], the contralateral neck for lateralized palatine tonsil cancer [55], the pathologically node-negative neck after selective neck dissection [56], after induction chemotherapy [57], in low risk nasopharynx carcinoma [58], and in HPV-associated oropharyngeal cancer [59]. Selectively decreasing the elective nodal radiation volumes has been associated with improved quality of life while not compromising disease control [55]. Importantly, decreased elective nodal radiation volumes also reduced iatrogenic immunosuppression [49]. In preclinical models, the presence of intact regional lymphatics appeared to be important for enhancing the immunogenicity of radiation therapy [15]. Future clinical trials of immunochemoradiotherapy may consider reducing or eliminating the use of elective nodal radiation to optimize the efficacy of the immunotherapy component of the treatment. To our knowledge, there is no ongoing Phase III study specifically designed to test the efficacy of immunochemoradiotherapy with the omission of elective nodal irradiation in locally advanced HNSCC. However, some ongoing trials such as RTOG 1216 (NCT01810913), which adds the anti-PD-L1 mAb atezolizumab to adjuvant CRT in surgically managed head and neck cancer, allow the selective omission of the elective nodal volume according to the discretion of the radiation oncologist, and future unplanned secondary analyses may discover interactions between radiation volumes and immunotherapy. It is important to note that modification of the radiotherapy volumes must be carefully selected to minimize the risk of compromising locoregional control.

Both PACIFIC and Checkmate 577 used immunotherapy after the completion of CRT instead of concurrent treatment. It is possible that allowing for time-sensitive recovery of the immune response prior to initiating checkpoint inhibitors might result in more favorable outcomes. Importantly, a recent Phase II study randomized 80 patients with locally advanced HNSCC receiving definitive CRT to either concurrent or sequential pembrolizumab [60]. Pembrolizumab was started one week prior to CRT and two weeks after CRT in the concurrent and sequential arms, respectively. The 1- and 2-year PFS for sequential pembrolizumab (89%) was numerically higher than that of for concurrent pembrolizumab (82% and 78%). Survival at 1 and 2 years was 94% for sequential and 82% and 78% for concurrent pembrolizumab. This study demonstrated that sequential treatment may be preferable to concurrent treatment in testing immunoradiotherapy treatment strategies. The ongoing Phase III IMvoke010 (NCT03452137) is testing a similar concept. Patients with locally advanced head and neck cancer are treated with definitive local therapy, which could include definitive CRT or surgery followed by RT or CRT as indicated. The patients are then randomized to adjuvant atezolizumab vs. a placebo. The ongoing EA3161 trial (NCT0811015) is a Phase III study in patients with intermediate-risk HPV-associated oropharyngeal cancer, which also tests adjuvant immunotherapy. After standard CRT, the patients are randomized to adjuvant nivolumab vs. observation. These studies will determine if sequential adjuvant immunotherapy after definitive local therapy will benefit patients with locally advanced HNSCC. Table 2 summarizes several ongoing trials that are testing the previously described concepts of immunotherapy with definitive CRT, in combination with surgery and adjuvant CRT, or in patients who are ineligible for concurrent cisplatin.

Finally, it is conceivable that better patient selection may allow for the use of immunotherapy and definitive radiation as an alternative to chemoradiation in patients with a more favorable prognosis, such as HPV-associated oropharyngeal cancer. KEYCHAIN (NCT03383094) is a Phase II study comparing standard CRT vs. RT plus pembrolizumab in patients with intermediate and high-risk p16-positive HNSCC. Additionally, NRG HN006 (NCT03952585) is a Phase II/III study in patients with low-risk HPV-associated oropharyngeal cancer, which randomizes patients to the standard of care (definitive CRT) vs. dose de-escalated definitive CRT (60 Gy) or dose de-escalated definitive RT (60 Gy) with nivolumab. The results of these two studies are awaited to determine if definitive radiation in combination with immunotherapy may find a role in some subgroups of patients with locally advanced HNSCC.

In the future, it will be important to identify the reason why definitive, concurrent immunochemoradiotherapy has not been a successful treatment strategy so far. Hypotheses currently include inadequate patient selection, the lack of biomarkers to predict the response, immunosuppression caused by concurrent therapy rather than sequential therapy, and immunosuppression caused by radiation of the elective lymph nodes. It will be important to conduct preclinical and clinical studies to evaluate these hypotheses, so that improved methods of incorporating immunotherapy with definitive chemoradiation can be tested in future.

## 6. Immunotherapy in the Multimodal Treatment of Recurrent and Metastatic Disease

The recurrent/metastatic HNSCC population represents a diverse group of patients with different primary disease sites, previous therapeutic approaches, patterns of recurrence, and disease burdens. The estimated recurrence rates in patients with locally advanced disease are estimated to be 40–50% for local recurrence and 20–30% for distant recurrence [61,62], while up to 10% of patients present with de novo metastatic disease [63]. The current NCCN guidelines recommend salvage surgery for resectable, recurrent HNSCC with recurrent or persistent disease following radiotherapy in the absence of distant metastatic disease. Radiation with concurrent systemic therapy is an NCCN Category 1 recommendation for recurrent or persistent locoregional disease in patients who have not received radiation for their initial therapy. Multimodal systemic combination therapy with or without radiotherapy for cytoreduction followed by local therapy is currently a Category 2B NCCN recommendation. The evolution of systemic therapy with the arrival of ICI could potentially lead to a shift in this treatment paradigm.

Historically, recurrent/metastatic HNSCC that are not amenable to local therapy have been treated with a non-curative systemic therapy combining platinum, 5-fluorouracil, and cetuximab, which was associated with an overall response rate of 36%, a high Grade 3–5 toxicity rate of 76%, and a median PFS of 5.6 months [3,64]. Following three landmark international studies, the systemic therapy approach in the first- and second-line recurrent/metastatic settings has shifted to the use of immunotherapy with the regulatory approvals of pembrolizumab and nivolumab. Pembrolizumab is currently approved as a single agent in the first-line management of recurrent/metastatic HNSCC expressing PD-L1 (combined positive score (CPS) ≥ 1), or in combination with chemotherapy in the first-line setting regardless of the expression of PD-L1. Both pembrolizumab and nivolumab were approved after progression on a regimen containing platinum.

Given the improvement in the outcomes from anti-PD1 mAb, recent clinical trials have investigated the novel use of radiotherapy as an immunotherapy sensitizer in recurrent/metastatic HNSCC. The underlying hypothesis is that hypofractionated radiotherapy delivered to a small number of metastatic lesions would cause immunogenic cell death, releasing the tumor antigens that act as an in situ vaccination enhancing the immune response to ICI. A classic example of this is the abscopal effect, which occurs when a tumor that was not treated with radiation regresses because of the immune response to another lesion that was treated with radiation [65]. Although it has been occasionally observed, the abscopal effect is rare and, to date, cannot be reliably predicted or induced in any malignancy. Preclinical studies have suggested that the immune response may depend on the radiation fraction size. A mouse tumor model treated with an anti-CTLA-4 antibody and radiotherapy observed abscopal effects after fractionated radiation of 8 Gy × 3 fractions and 6 Gy × 5 fractions, but not after a single fraction of 20 Gy [66]. In this setting, the 8 Gy × 3 fraction radiation treatment was the more effective regimen for inducing the immune response. In clinical practice, stereotactic body radiation (SBRT) is defined as hypofractionated radiation (≥5 Gy per fraction, treatment delivered in one to five fractions) delivered with conformal radiation fields, while hypofractionated radiation typically is considered to be ≥2.5 Gy per fraction. Preliminary clinical trials have typically used SBRT in combination with one of several ICI agents, including avelumab, pembrolizumab, durvalumab, or nivolumab. Multiple single-arm open-label trials are evaluating various combinations of hypofractionated radiotherapy and ICI in recurrent/metastatic HNSCC, although few have been reported (NCT03844763, NCT03474497, and NCT03386357). The initial analysis of a Phase I/II trial (NCT03283605) of SBRT combined with dual anti-PD-L1 and anti-CTLA-4 mAbs (durvalumab and tremelimumab) showed a promising signal with a response in 4 out of 7 patients with target lesions untreated by SBRT, and in 9 out of 14 patients if SBRT-treated lesions were included in the RECIST analysis [67]. However, a single-institution Phase II trial on metastatic HNSCC evaluating nivolumab vs. nivolumab plus stereotactic radiation (9 Gy × 3) to a single metastatic lesion found no significant difference in PFS or OS [68]. Contrary to the hypothesis, they found no evidence that this combination induced the abscopal effect. A larger Phase III trial currently underway, PembroMetaRT (NCT04747054), is evaluating the benefit of adding hypofractionated locoregional radiotherapy (54 Gy in 18 fractions) to first-line pembrolizumab or pembrolizumab plus chemotherapy in patients with newly diagnosed HNSCC with synchronous metastasis.

ICIs are also under investigation in the re-irradiation setting for patients with locoregionally recurrent disease. REPORT (NCT03317327), a Phase I/II trial adding nivolumab to hyperfractionated re-irradiation (1.5 Gy twice daily to 60 Gy) for recurrent or secondary primary HNSCC, is currently enrolling patients, with the primary results on safety expected to be reported at the end of 2023. Secondary endpoints will include PFS and the duration of the response. A similar trial with pembrolizumab and hyperfractionated re-irradiation (1.2 Gy twice daily to 60 Gy) is also underway (NCT02289209). The rationale for hyperfractionated radiation (multiple small fractions per day) is based on the assumption that a small fractionation size might decrease late toxicity, which is important in the re-treatment setting [69]. The interaction between hyperfractionated radiation and ICIs is still unclear. Re-irradiation with SBRT and ICI is also under investigation. KEYSTROKE (NCT03546582) is evaluating re-irradiation with SBRT over 2 weeks with or without sequential pembrolizumab in patients with locoregionally recurrent or second primary HNSCC. A Phase I re-irradiation study (NCT05526924) will assess the safety, tolerability, and maximum tolerated dose of tislelizumab, an anti-PD1 mAb, in combination with pamiparib, a PARP inhibitor, added to hyperfractionated CRT using 5-fluorouracil and hydroxyurea. Brachytherapy, which delivers radiation through implanted radioactive isotopes rather than via an external beam, might also be effective as an in situ tumor vaccine while sparing the surrounding immune cells from radiation [70]. A single-institution Phase I/II trial (NCT04340258) is evaluating pembrolizumab combined with cesium 131 brachytherapy and salvage surgery for the treatment of recurrent head and neck cancer.

The role of immunotherapy in the multimodal treatment of recurrent and metastatic disease is therefore very complex, and many strategies are currently being tested in early studies. Going forward, the main question remains whether a multimodal regimen combining immunotherapy and radiation can be identified that is promising enough to advance to a Phase III trial against the standard first-line treatment of immunotherapy or immunochemotherapy. Additionally, the question remains whether the abscopal effect can be reliably induced by the combination of SBRT and immunotherapy in metastatic HNSCC.

## 7. Conclusions

Immune checkpoint inhibitors have dramatically changed the treatment landscape for recurrent/metastatic HNSCC. This success has driven investigations into treatment strategies incorporating immunotherapy earlier into the multimodal curative-intent or salvage treatment of both locally advanced and recurrent/metastatic HNSCC. Definitive immunochemoradiotherapy has not been shown to be more effective than definitive chemoradiotherapy alone in unselected patients with locally advanced HNSCC, but trials are still ongoing to see if certain subsets may benefit. Clinical trials are ongoing to investigate the role of immunotherapy in the neoadjuvant treatment of surgically resectable HNSCC and as part of combination treatment in recurrent/metastatic HNSCC. The results of these studies are eagerly awaited to improve patient outcomes in this challenging disease.

## Figures and Tables

**Table 1 cancers-15-00672-t001:** Selected Phase I/II clinical trials of neoadjuvant ICI prior to surgery for locally advanced HNSCC.

NCT Number	Study Name	Eligible Disease	Description	Outcome
**Monotherapy Neoadjuvant +/− Adjuvant after Surgery**				
NCT02488759	CheckMate-358	Stage III–IV resectable HNSCC (HPV-agnostic)	N = 52 Nivolumab (240 mg IV) on Days 1 and 15, surgery planned day 29; no surgical delay > 4 weeks.	Nivolumab safe; pathologic regression of 3.5% of tumors for HPV+ and 5.9% for HPV− [25]
NCT02296684	Uppaluri et al., 2020 (WashU and Harvard)	Stage III–IV resectable HNSCC (not HPV or sinonasal)	N = 36 Pembrolizumab (200 mg IV), one dose followed by surgery 2–3 weeks later; high-risk pathology also received pembrolizumab after adjuvant CRT; no surgical delay due to AE	Pembrolizumab safe; pathologic regression of tumor by 44% overall (>50% seen in 22%); one-year relapse, 16.7% [26]
NCT02641093	Wise-Draper et al., 2022 (Cincinnati)	Stage III–IV resectable HNSCC; T3/T4 or >2 LN/ENE. (not HPV+ oropharynx or nasopharynx)	N = 75 Pembrolizumab (200 mg IV), one dose followed by surgery 2–3 weeks later; high-risk pathology also received pembrolizumab along with adjuvant CRT; no surgical delay due to AE	Difference seen in low-risk (96%) vs. high-risk (69%) groups DFS at one year. The pathologic response was predictive of DFS. The timing of ICI may lead to lower DFS than prior studies [27]
**Combinated Anti-PD-L1/Anti-CTLA-4 Neoadjuvant**				
NCT03003637	IMCISION	Stage II–IV resectable HNSCC, some recurrent	N = 32 (26 combination) Nivolumab (240 mg IV), ×2 in Weeks 1 and 3, plus ipilimumab (1 mg/kg in Week 1 only; no surgical delays	Combination tx induced a 35% major pathological response at the primary tumor site; SAE, 38%; Grade 3/4 [28]
NCT02919683	Schoenfeld et al., 2020 (Harvard)	Stage T2-4b/N+ oral cavity SCC	N = 29 Nivolumab neoadjuvant +/− ipilimumab (15 patients); no surgical delays. Nivolumab given in Weeks 1 and 3; ipilimumab was given in Week 1 only (1 mg/kg)	For the monotherapy or combined arms: 53% downstaging vs. 69%; pathological response of 53% vs. 73%; mPR/CR in 1 vs. 3 patients [29]
NCT03144778	CIAO	Stage II–IVA OPSCC, new or recurrent, surgically resectable	N = 28 Neoadjuvant durvalumab +/− tremelimumab; endpoint to measure CD8+ TIL density	Combined therapy did not show increased TIL; the safety profile was confirmed; 29% mPR (<10% of the viable tumor left) [30]
**Immunotherapy with Surgery and Chemoradiation**				
NCT03247712	Neoadjuvant Immuno-Radiotherapy Trial (Oregon)	Stage I–III p16+ or Stage III–IVA p16− HNSCC; no prior treatment	N = 21 Neoadjuvant SBRT +/− nivolumab; no delays to surgery; safety profile met. Adjuvant nivolumab planned ×3 months postoperatively	PORT eliminated a patient in 20/21 due to a favorable pathologic response; 86% mPR; 90% downstaging (mostly p16+) [31]
NCT03342911	Zinner et al., 2020 (Thomas Jefferson University)	Stage II–IV HNSCC, resectable with post-operative XRT planned	N = 27 Neoadjuvant carboplatin (C), paclitaxel (P), and nivolumab (N); surgery in Week 8; safety profile met	Pending final results; the initial results showed pCR 11/26 (42%) with 69% mPR vs. 65% in HPV+ vs. HPV− tumors [32]
NCT05459129	Morpheus	LAHNSCC, resectable	Planning to enroll 180 patients to 4 arms: atezolizumab +/− tiragolumab and neoajuvant SBRT vs. carboplatin/paclitaxel arms	Pending, currently enrolling

HPV—Human papillomavirus, CRT—chemoradiation, AE—adverse events, LN—lymph nodes, ENE—extranodal extension, DFS—disease free survival, ICI—immune checkpoint inhibitor, SAE—severe adverse event, SCC—squamous cell carcinoma, OPSCC—oropharyngeal squamous cell carcinoma, CD8—cluster of differentiation 8, TIL—tumor infiltrating lymphocytes, PORT—postoperative radiotherapy, LAHNSCC—locally advanced head and neck squamous cell carcinoma, SBRT—stereotactic body radiotherapy.

**Table 2 cancers-15-00672-t002:** Selected Phase III clinical trials of immunochemoradiation in locally advanced head and neck cancer.

NCT Number	Study Name	Eligible Disease	Description	Outcome
**Concurrent Immunotherapy with Definitive Chemoradiation**				
NCT02952586	JAVELIN HN 100	LAHNSCC	1: SOC definitive CRT 2: 2SOC definitive CRT with concurrent and adjuvant avelumab	Primary endpoint of improved PFS not reached [41]
NCT03040999	KEYNOTE-412	LAHNSCC	1: SOC definitive CRT 2: 2SOC definitive CRT with concurrent and adjuvant pembrolizumab	Primary endpoint of improved EFS not reached [42]
NCT03952585	NRG-HN005	Early-stage p16-positive oropharyngeal cancer	1: SOC definitive CRT (70 Gy) 2: Low-dose definitive CRT (60 Gy) 3: Low-dose definitive RT (60 Gy) with concurrent nivolumab	Pending
**Sequential Immunotherapy after Definitive Chemoradiation**				
NCT0811015	EA3161	Intermediate-risk p16-positive oropharyngeal cancer	1: SOC definitive CRT 2: SOC definitive CRT with adjuvant nivolumab	Pending
**Immunotherapy with Surgery and Chemoradiation**				
NCT01810913	RTOG 1216	Resected p16-negative LAHNSCC	1: SOC surgery and adjuvant CRT (cisplatin) 2: SOC surgery and adjuvant CRT (docetaxel and cetuximab) 3: SOC surgery and adjuvant CRT (cisplatin) with atezolizumab. Atezolizumab is concurrent and sequential with CRT	Pending
NCT03576417	NIVOPOSTOP	Resected LAHNSCC	1: SOC surgery and adjuvant CRT 2: SOC surgery and adjuvant CRT with nivolumab. The nivolumab is concurrent with and sequential to the CRT	Pending
NCT03452137	IMvoke010	Definitively treated LAHNSCC (CRT or surgery as the definitive local therapy)	Definitive local therapy followed by: 1: Placebo 2: Atezolizumab	Pending
**Immunotherapy as a Neoadjuvant Therapy Prior to Surgery and also with Adjuvant CRT**				
NCT03765918	MK-3475-689	Resectable LAHNSCC	1: SOC surgery and adjuvant CRT 2: Neoadjuvant pembrolizumab, surgery, and adjuvant CRT with pembrolizumab. Pembrolizumab is a neoadjuvant prior to surgery, concurrent with adjuvant CRT and sequential following CRT	Pending
NCT03700905	IMSTAR-HN	Resectable LAHNSCC	1: SOC surgery and adjuvant CRT 2: Neoadjuvant nivolumab, surgery, and adjuvant CRT, followed by sequential nivolumab or adjuvant nivolumab and ipilimumab	Pending
**Immunotherapy for Patients Ineligible for Cisplatin**				
NCT03258554	NRG-HN004	LAHNSCC, cisplatin ineligible	1: RT with cetuximab 2: RT with durvalumab, given concurrent and adjuvant	Primary endpoint of improved PFS not reached [45]
NCT02999087	REACH	LAHNSCC, cisplatin eligible and -ineligible	1: SOC CRT (cisplatin) for cisplatin-eligible patients 2: RT with cetuximab in cisplatin-ineligible patients 3: RT with concurrent cetuximab and avelumab, followed by adjuvant avelumab	Primary endpoint of improved PFS not reached for cisplatin-eligible and -ineligible patients [43]

SOC, standard of care; CRT, chemoradiation; PFS, progression-free survival; EFS, event-free survival; LAHNSCC, locally advanced head and neck squamous cell carcinoma; RT, radiation therapy.

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
