# Peer review of "Integrating Immunotherapy into Multimodal Treatment of Head and Neck Cancer"

_cancers, 2023, doi:10.3390/cancers15030672_

Round 1

Reviewer 1 Report

Please describe a little clear the conclusions 

Author Response

Reviewer 1

Please describe a little clear the conclusions

Author’s response:  Thank you, we have clarified the conclusion section to add clear conclusion items related to the text and to improve readability.

Reviewer 2 Report

It is a very interesting work in which the possibilities of treating head and neck tumors with immunotherapy are evaluated.

Also significant is the analysis of neoadjuvant therapy protocols that could offer new treatment opportunities.

The paper is complete and exhaustive and can be accepted in the present form.

Author Response

Reviewer 2

It is a very interesting work in which the possibilities of treating head and neck tumors with immunotherapy are evaluated.

Also significant is the analysis of neoadjuvant therapy protocols that could offer new treatment opportunities.

The paper is complete and exhaustive and can be accepted in the present form.

Author’s response:  Thank you

Reviewer 3 Report

In the treatment of head and neck cancer, Rao et al. concentrated on immunotherapy. They provided us with a rather in-depth overview of the multimodality treatment of head and neck cancer in this section. Overall, this manuscript's timeliness and accuracy are respectable for a review. I do advise the authors to make the majority of the changes to the paper. Here are some significant remarks: 

1.     Some aspects of this systematic review, which was centered on immunotherapy, are not well supported by the literature. The discussion's breadth and depth, meantime, are lacking. For instance, only 4 studies were cited in the section on "The Immune Microenvironment in HNSCC." It is challenging to paint a comprehensive picture of the immunological microenvironment in HNSCC in this section. 

2.     It is not essential to review the immunotherapy's past. 

3.     Most notably, although reviewing the therapeutic impact of immunotherapy in head and neck cancer with various clinical characteristics from five angles, the manuscript ultimately lacks substance. The author doesn't appear to have discussed the role of immunotherapy in the manuscript; instead, it appears to merely evaluate research on immunotherapy for head and neck squamous cell carcinoma. Through this study, it is difficult to comprehend immunotherapy's features, benefits, and drawbacks in managing head and neck squamous cell carcinoma. Additionally, there is a paucity of comparisons with other cancer treatment concepts and consequences. More importantly, the review did not suggest the best course of action for future research.

In addition, the list below includes a few small recommendations and inquiries. 

1.     It is necessary to standardize reference citation styles. 

2.     The data in the tables should have a list of their sources.

Author Response

Reviewer 3

In the treatment of head and neck cancer, Rao et al. concentrated on immunotherapy. They provided us with a rather in-depth overview of the multimodality treatment of head and neck cancer in this section. Overall, this manuscript's timeliness and accuracy are respectable for a review. I do advise the authors to make the majority of the changes to the paper. Here are some significant remarks:

  1. Some aspects of this systematic review, which was centered on immunotherapy, are not well supported by the literature. The discussion's breadth and depth, meantime, are lacking. For instance, only 4 studies were cited in the section on "The Immune Microenvironment in HNSCC." It is challenging to paint a comprehensive picture of the immunological microenvironment in HNSCC in this section.

Author’s response: We have combined our response to #1 and #3; please see #3 below. 

  1. It is not essential to review the immunotherapy's past.

Author’s response:  Thank you for this comment, this part of the manuscript has been shortened to improve readability for readers who are not interested in the historical aspects of immunotherapy.  The history is now very briefly reviewed in a single paragraph instead of several.

  1. Most notably, although reviewing the therapeutic impact of immunotherapy in head and neck cancer with various clinical characteristics from five angles, the manuscript ultimately lacks substance. The author doesn't appear to have discussed the role of immunotherapy in the manuscript; instead, it appears to merely evaluate research on immunotherapy for head and neck squamous cell carcinoma. Through this study, it is difficult to comprehend immunotherapy's features, benefits, and drawbacks in managing head and neck squamous cell carcinoma. Additionally, there is a paucity of comparisons with other cancer treatment concepts and consequences. More importantly, the review did not suggest the best course of action for future research.

Author’s response:  Thank you for this important criticism of the manuscript.  The section on immunotherapy and the immune microenvironment has been dramatically expanded to improve the manuscript and to improve the description of the tumor microenvironment, the immune environment, and interactions of treatment with this environment.  Regarding comparisons to other treatments, we argue that some of these comparisons are already present in the manuscript in that the clinical trials discussed compare with the standard of care (either surgery or chemoradiation) which are the standard (and alternative) treatments to immunotherapy in these clinical situations.  These are now emphasized in the edits.  Additionally, to address the remarks, we have provided additional information on potential courses of action of future research in each of the major sections of the manuscript.

In addition, the list below includes a few small recommendations and inquiries.

  1. It is necessary to standardize reference citation styles.

Author’s response:  Thank you, we have made sure that the citation styles match the requested format for the journal Cancers.

  1. The data in the tables should have a list of their sources.

Author’s response:  The tables have been updated to include citations, where there is published data.  For clinical trials without any published data, no citation is added and interested readers will find the NCT number to be sufficient to find publicly available information.

Round 2

Reviewer 3 Report

The authors responded to most of my concerns. I have no further questions.